# The Knock-Down of the Chloroquine Resistance Transporter PfCRT Is Linked to Oligopeptide Handling in *Plasmodium falciparum*

Cecilia P. Sanchez,[a] Erin D. T. Manson,[b] Sonia Moliner Cubel,[a] Luis Mandel,[b] Stefan K. Weidt,[c] Michael P. Barrett,[c,d] (ID) Michael Lanzer[a]

[a]Center of Infectious Diseases, Parasitology, Universitätsklinikum Heidelberg, Heidelberg, Germany
[b]NTT DATA Deutschland, München, Germany
[c]Glasgow Polyomics, University of Glasgow, Wolfson Wohl Cancer Research Centre, Glasgow, United Kingdom
[d]The Wellcome Centre for Integrative Parasitology, Institute for Infection, Immunity and Inflammation, College of Medical, Veterinary and Life Sciences, University of Glasgow, Glasgow, United Kingdom

**ABSTRACT** The chloroquine resistance transporter, PfCRT, is an essential factor during intraerythrocytic development of the human malaria parasite *Plasmodium falciparum*. PfCRT resides at the digestive vacuole of the parasite, where hemoglobin taken up by the parasite from its host cell is degraded. PfCRT can acquire several mutations that render PfCRT a drug transporting system expelling compounds targeting hemoglobin degradation from the digestive vacuole. The non-drug related function of PfCRT is less clear, although a recent study has suggested a role in oligopeptide transport based on studies conducted in a heterologous expression system. The uncertainty about the natural function of PfCRT is partly due to a lack of a null mutant and a dearth of functional assays in the parasite. Here, we report on the generation of a conditional PfCRT knock-down mutant in *P. falciparum*. The mutant accumulated oligopeptides 2 to at least 8 residues in length under knock-down conditions, as shown by comparative global metabolomics. The accumulated oligopeptides were structurally diverse, had an isoelectric point between 4.0 and 5.4 and were electrically neutral or carried a single charge at the digestive vacuolar pH of 5.2. Fluorescently labeled dipeptides and live cell imaging identified the digestive vacuole as the compartment where oligopeptides accumulated. Our findings suggest a function of PfCRT in oligopeptide transport across the digestive vacuolar membrane in *P. falciparum* and associated with it a role in nutrient acquisition and the maintenance of the colloid osmotic balance.

**IMPORTANCE** The chloroquine resistance transporter, PfCRT, is important for the survival of the human malaria parasite *Plasmodium falciparum*. It increases the tolerance to many antimalarial drugs, and it is essential for the development of the parasite within red blood cells. While we understand the role of PfCRT in drug resistance in ever increasing detail, the non-drug resistance functions are still debated. Identifying the natural substrate of PfCRT has been hampered by a paucity of functional assays to test putative substrates in the parasite system and the absence of a parasite mutant deficient for the PfCRT encoding gene. By generating a conditional PfCRT knock-down mutant, together with comparative metabolomics and uptake studies using fluorescently labeled oligopeptides, we could show that PfCRT is an oligopeptide transporter. The oligopeptides were structurally diverse and were electrically neutral or carried a single charge. Our data support a function of PfCRT in oligopeptide transport.

**KEYWORDS** PfCRT, oligopeptides, metabolomics, malaria, *P. falciparum*, membrane transport, substrate

**Ad Hoc Peer Reviewer** (ID) Benjamin Liffner, Indiana University School of Medicine

Address correspondence to Michael Lanzer, michael.lanzer@med.uni-heidelberg.de.

The authors declare no conflict of interest.

The human malaria parasite *Plasmodium falciparum* is an obligatory intracellular parasite that initially infects hepatocytes after transmission by a blood feeding *Anopheles* mosquito before the parasite changes its host cell specificity to replicate within erythrocytes. Much of the pathology associated with *falciparum* malaria relates to the blood stages, whereas the liver stages remain asymptomatic. During intra-erythrocytic development, the parasite ingests up to 80% of the hemoglobin present in the red blood cell in a specialized proteolytic organelle, termed the digestive vacuole, to sustain its own anabolic activity and maintain the colloid-osmotic balance of the infected erythrocyte (1–3). The engulfed hemoglobin is enzymatically degraded, and the liberated heme mineralizes to inert hemozoin in the digestive vacuole (4, 5). Several classes of antimalarial drugs, including cinchona alkaloids, quinoline-like drugs and sesquiterpene lactones, target hemoglobin degradation or heme detoxification (6, 7). To evade the activity of these drugs, the parasite has evolved resistance mechanisms that involve altered drug handling at the digestive vacuolar membrane or reduced hemoglobin endocytosis (8–10).

A prominent drug transporting system of the digestive vacuolar membrane is the chloroquine resistance transporter, PfCRT (11). Polymorphisms in PfCRT confer resistance to the former first line antimalarial, chloroquine, and they can modulate the responsiveness to quinine and quinine-like drugs but also to structurally unrelated compounds (12, 13). PfCRT belongs to the drug/metabolite carrier superfamily (14). It features 10 transmembrane domains and an internal 2-fold pseudo-symmetry, and it can carry 4 to 10 drug resistance-associated polymorphisms depending on geographic origin and local drug history (13–15). These polymorphisms can incur a fitness cost on the parasite, alter hemoglobin metabolism, and cause the digestive vacuole to swell (16–21), suggesting that PfCRT exerts an important physiological function during intra-erythrocytic development. This conclusion is further supported by the finding that attempts to knockout *pfcrt* have thus far failed and that inhibiting PfCRT with targeted compounds kills the parasite (22, 23).

Although the drug-transporting function of PfCRT is understood in ever greater detail, the natural non-drug related function has remained unsettled, despite—or because of—a long list of proposed physiological roles. For example, previous studies have implicated PfCRT in pH homoeostasis (24), chloride flux (25), exchange of ferrous and ferric iron (26), glutathione transport (27), and polyspecific transport of cations (28). In a recent development, it has been reported that PfCRT can transport oligopeptides 4 to 11 residues in length depending upon protons and in co-transport with a hitherto unknown substance in the heterologous *Xenopus laevis* oocyte system (29).

Identifying the physiological function of PfCRT has been hampered by a dearth of functional assays to test the various hypotheses in the parasite system. Moreover, a PfCRT null or a knock-down mutant is not yet available and, accordingly, the loss of gene function on the physiology of the parasite has not yet been investigated. Here, we report the successful generation of a conditional *pfcrt* knock-down mutation in the chloroquine resistant parasite line Dd2. We show that the mutant accumulates oligopeptides in the digestive vacuole under knock-down conditions, as suggested by comparative metabolomics and uptake studies using fluorescently labeled oligopeptides.

## RESULTS

***Pfcrt* knock-down mutant reverts to drug sensitivity and bears a fitness cost.**
Fig. 1 outlines the strategy to generate a conditional *pfcrt* knock-down mutant by inserting the glucosamine 6-phosphate inducible glms ribozyme sequence, together with a hemagglutinin (HA) tag, in the 3′ untranslated region of the *pfcrt* locus of the chloroquine resistant parasite line Dd2, via CRISPR/Cas9 genome editing technology (30, 31) (Fig. 1A). Two independent clones, termed E10 and G7, were obtained. The integration event was confirmed by PCR and sequencing analyses of genomic DNA and mRNA (Fig. 1B). Induction of the knock-down phenotype for 3 days with 1 mM glucosamine (GlcN) reduced steady state PfCRT protein levels by 75%, compared with

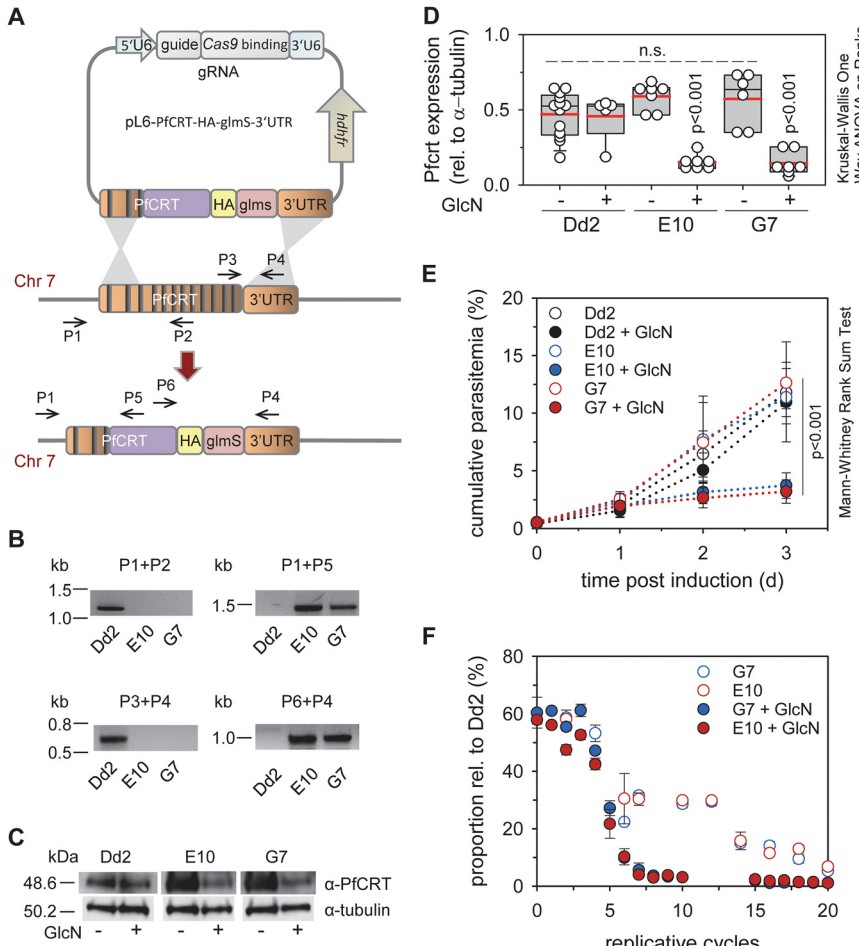

**FIG 1** Generation and analysis of conditional PfCRT knock-down mutants. (A) Cartoon depicting the transfection strategy. The genomic *pfcrt* locus (dark orange), the recodonized *pfcrt* fragment (purple), the hemagglutinin tag (HA, yellow) and the glmS ribozyme sequence (pink) are shown. The positions of relevant primers are indicated (Table S10). Introns are represented as vertical black lines. (B) The integration event was verified by PCR, using genomic DNA from the resulting mutants E10 and G7 and the parental Dd2 line. Size markers are indicated in kilo base pairs (kb). (C) Western analysis of total protein derived from trophozoites of Dd2 and the PfCRT knock-down mutants in the presence and absence of glucosamine (1 mM for 3 days), using a guinea pig anti-PfCRT antiserum and as a loading control a mouse monoclonal anti-tubulin antibody and as secondary antibodies a goat anti-guinea pig POD antibody and a donkey anti-mouse POD antibody, respectively. A size standard is indicated in kDa. (D) Independent Western analyses were quantified and PfCRT specific luminescence signals were normalized to those of tubulin. Each symbol represents an independent biological replicate. The statistical test used for data analysis is indicated in this and the following figures in the graph (box plot analysis, see Materials and Methods). (E) Asexual intraerythrocytic proliferation. Proliferation of the conditional *pfcrt* knock-down mutants, E10 and G7, and the parental strain Dd2 was determined over 72 h in the presence and absence of 1 mM GlcN. Time point 0 indicates the time point at which the cell cultures (the ring stages) were split and one aliquot each was incubated in the presence of GlcN. The other aliquots served as the untreated controls. The cumulative parasitemia is the sum of the parasitemia in the culture after considering splitting and dilution factors. The means ± SEM of four independent experiments are shown. (F) Fitness test. Mixed cultures of Dd2 and each of the PfCRT knock-down mutants were maintained in the presence or absence of 1 mM glucosamine for 20 replicative cycles. The allelic proportions were measured by pyrosequencing. The data represent the means ± SEM of three independent biological replicates.

untreated controls, as shown by semi-quantitative Western analysis (Fig. 1C and D). Furthermore, mutants displayed a lower replication rate in the knock-down state (Fig. 1E), and they bore a fitness cost both in the presence and absence of GlcN in relation to the parental line Dd2 (Fig. 1F). As expected, knock-down of PfCRT sensitized the parasite to choroquine and desethyl amodiaquine (the active metabolite of amodiaquine), as indicated by low IC$_{50}$-values and high drug accumulation ratios (Fig. 2). In

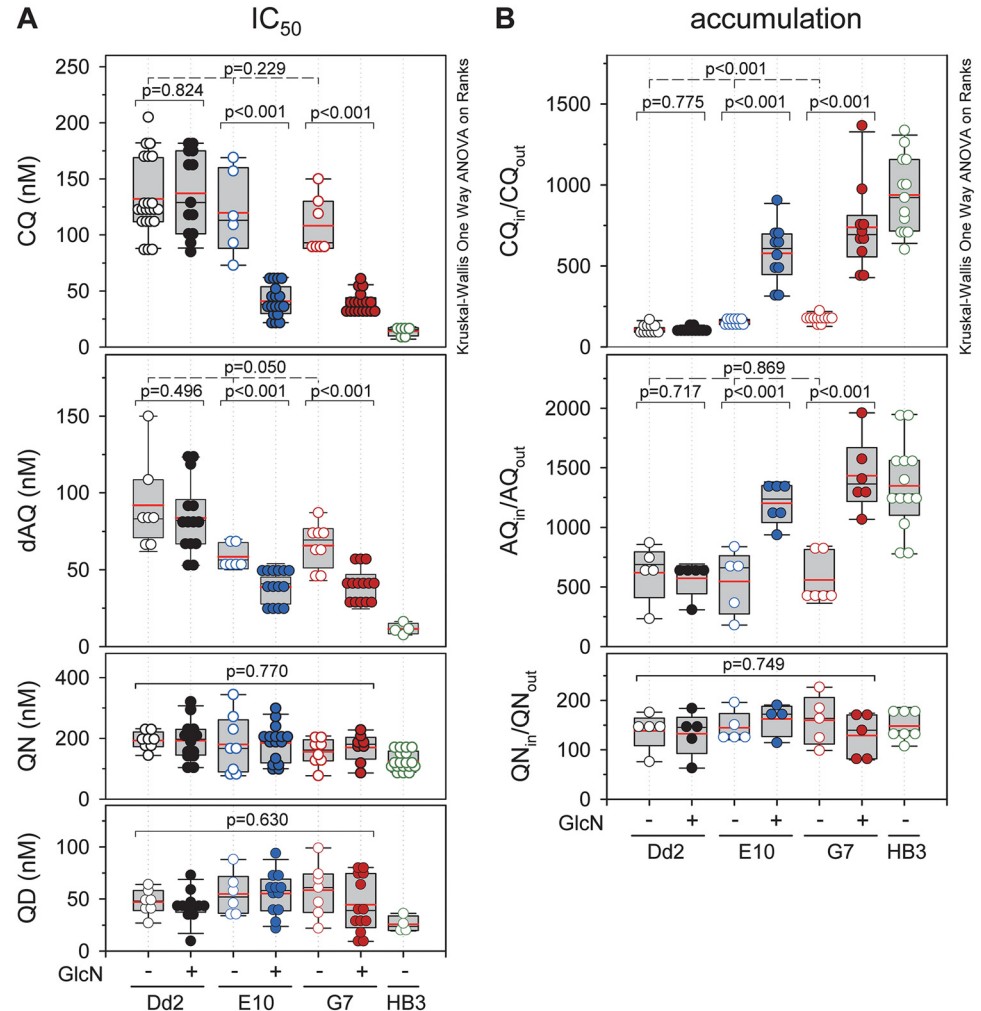

**FIG 2** Drug responses of the PfCRT knock-down mutants. (A) Growth inhibition assay. Synchronized parasite cultures at the ring stage were incubated with increasing concentrations of the drugs indicated for 72 h. Parasite viability was expressed in terms of the half maximal inhibitory drug concentration ($IC_{50}$). (B) Drug accumulation assay. Purified parasites at the trophozoite stage were incubated with 40 nM radiolabeled drugs for 20 min before the amount of radiolabel taken up by the cell ($Drug_{in}$) and the amount remaining in the buffer ($Drug_{out}$) were determined. Drug uptake was expressed in terms of the ratio of the intracellular versus extracellular drug concentration ($Drug_{in}/Drug_{out}$). The following drugs were used chloroquine (CQ), amodiaquine (AQ), monodesethyl amodiaquine (dAQ), quinine (QN) and quinidine (QD). Data are shown for the PfCRTknock-down mutants E10 and G7, the parental line Dd2 and the drug sensitive line HB3. Each symbol represents an independent biological replicate.

comparison, no significant effects on quinine and quinidine responsiveness were observed (Fig. 2).

**Pfcrt knock-down results in the accumulation of oligopeptides.** To assess the effect of the PfCRT knock-down on the physiology of the parasite, we conducted an untargeted and unbiased global metabolomics analysis, using highly synchronized parasite cultures of Dd2, E10 and G7 20 to 26 h post invasion. To provide a kinetic context, we investigated untreated cells and cells exposed to 1 mM GlcN for 1 and 3 days (Fig. 3). A total of 807 metabolites were identified and relative quantities compared between knock-down and wild type cells (Tables S1 and S2 in the supplemental material). Two hundred twenty-one and 109 metabolites were significantly more abundant ($P < 0.01$; LOD score $> 1.08$; fold change $> 2$) in 1 and 3 day treated knock-down mutants as compared with the control group (Tables S3 and S4). The data were highly consistent between the two parasite mutants investigated - 87% (192 of 221 and 95 of 109) in both treatment groups and were subsequently pooled for further analysis.

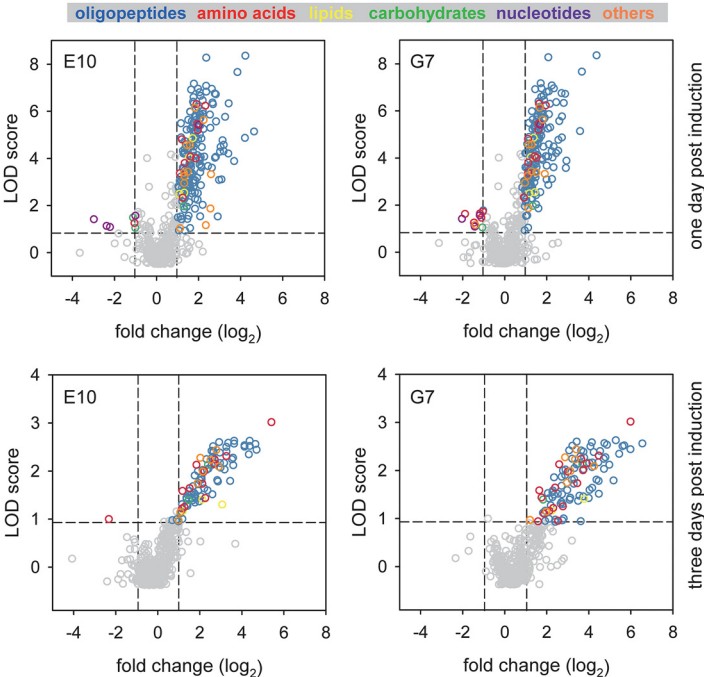

**FIG 3** Comparative global metabolomic profiling of PfCRT knock-down mutants. Volcano plots depict the fold change in metabolites between glucosamine treated (for 1 or 3 days) and untreated PfCRT knock-down mutants E10 and G7. Each dot represents a metabolite. Gray dots indicate metabolites that were not significantly associated with the induced PfCRT knock-down phenotype ($P > 0.01$, dashed horizontal line, or $-1 <$ fold change $(\log_2) < 1$, dashed vertical lines). Metabolite in the upper left and upper right quadrants indicate metabolites that were significantly down- and upregulated in the glucosamine treated cells, respectively. Colors indicate different metabolite classes.

Importantly, 83% (183 of 221) and 71% (77 of 109) of the metabolites linked with the PfCRT knock-down phenotype were oligopeptides, of which 62 were present in both treatment groups (Fig. 4A). The second and third most frequent metabolite classes were amino acids and their metabolites (6.8% and 17.4%, respectively) followed by lipids (1.8% and 2.8%, respectively) (Fig. 4A).

The initially identified oligopeptides were 2 to 4 amino acids in length (Fig. 4B). Oligopeptides of more than 4 residues could not be detected in the unbiased metabolomics approach due to constraints on the database size of the IDEOM analysis software, which is confined to peptides of 4 amino acids or smaller. Moreover, the analysis tools did not discriminate between different isomeric forms of a given peptide, which, in turn, suggests a much larger PfCRT-associated peptidome than inferred from the number of oligopeptide entries in the IDEOM output file.

To explore the PfCRT-associated peptidome landscape and the putative origin of the oligopeptides, we searched the erythrocyte proteome for matches to all possible isomeric permutations of the oligopeptides identified. 655 and 266 unique oligopeptides were identified in the 1- and 3-day PfCRT knock-down groups. Importantly, all of these oligopeptides could be related to 1 of the 12 most abundant red blood cell proteins, including $\alpha$- and ß-globin (Tables S5 and S6 in the supplemental material), suggesting that they are degradation products of these proteins.

We next complemented the unbiased metabolomics analysis with a targeted approach and matched the masses from the metabolomics data sets to putative larger peptides from the 12 key red blood cell proteins that had been identified as contributing to the smaller peptide pool, and numerous other peptides of larger size (but falling within the 1200 mass window ceiling set on the mass spectrometry). Three hundred eighty-seven and 174 additional putative oligopeptides ranging from five to eight residues could be identified in the two treatment groups, which were significantly associated with PfCRT downregulation (Fig. 4B and Table S7 in the supplemental material).

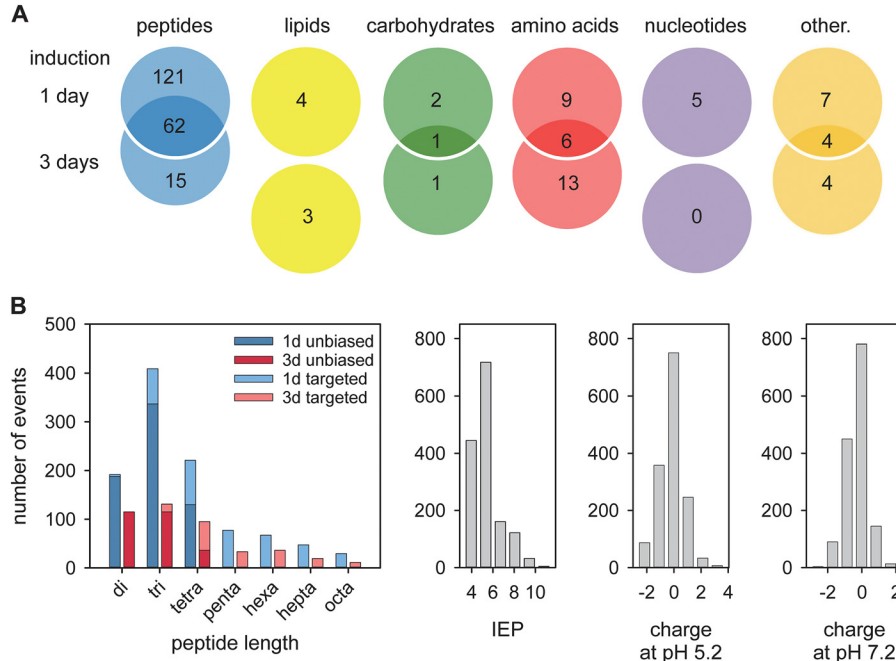

**FIG 4** Analysis of metabolites and oligopeptides associated with the induced PfCRT knock down. (A) Metabolites were categorized, and each category was analyzed using a Venn diagram illustrating the number of metabolites that were distinct and common between the 1- and 3-day glucosamine treated PfCRT knock-down mutants. (B) Analysis of the oligopeptides associated with the induced PfCRT knock-down phenotype according to peptide size (color codes indicate the 1- and 2-day induction regimen and the unbiased and targeted metabolomics approaches), isoelectric point (IEP ranging from 3.27 to 11.49) and charge distribution at pH 5.2 (ranging from -2.70 to 3.73) and 7.2 (ranging from -3.00 to 2.11).

The loss of the transporter thus also contributes to an accumulation of larger peptides, indicating that it recapitulates *in situ* the general peptide transporting activity of the transporter expressed in *Xenopus laevis* oocytes (29).

An analysis of the combined peptide pools revealed that most of the oligopeptides were two to four residues long, had isoelectric points between 4.0 and 5.4 (Fig. 4B) and were electrically neutral or carried a single positive or negative charge at the pH of 5.2 of the digestive vacuole or at the pH of 7.2 of the parasite cytoplasm (Fig. 4B).

**Fluorescently labeled di-peptides accumulate in the digestive vacuole in PfCRT knock-down mutants.** A microscopic inspection of Giemsa-stained blood smears revealed enlarged and translucent digestive vacuoles in the PfCRT knock-down mutants both in the presence and absence of GlcN (Fig. 5A). Quantification of the digestive vacuolar volume, based on live-cell imaging of infected erythrocytes (at the trophozoite stage 22 to 28 h post invasion) stained with the digestive vacuolar marker, acridine orange, confirmed this result (Fig. 5B). For Dd2, the volumes of the digestive vacuole were $4.6 \pm 0.2 \ \mu m^3$ and $4.7 \pm 0.2 \ \mu m^3$ in the presence and absence of GlcN, respectively, consistent with previous reports (32). In comparison, the PfCRT knock-down mutants had significant larger digestive vacuole volumes both in the absence ($5.8 \pm 0.2 \ \mu m^3$ and $5.4 \pm 0.2 \ \mu m^3$ for E10 and G7) and the presence ($7.1 \pm 0.4 \ \mu m^3$ and $7.6 \pm 0.3 \ \mu m^3$) of GlcN ($P \leq 0.01$ Mann-Whitney Rank Sum Test and Kruskal-Wallis one-way analysis of variance on ranks). These observations led us to hypothesize that the oligopeptides accumulate in the digestive vacuole where they might cause osmotic stress. To test this hypothesis, we incubated parasite cultures with five different pools of custom-made fluorescently labeled di-peptides overnight (Fig. 6A and B and Table S8 in the supplemental material) and then quantified the specific fluorescence signal in the digestive vacuole in relation to the overall cellular fluorescence. This experimental strategy was inspired by previous studies demonstrating passage of oligopeptides across the erythrocyte plasma membrane via parasite-induced new permeation pathways followed by endocytosis together with host cell cytoplasm and subsequent

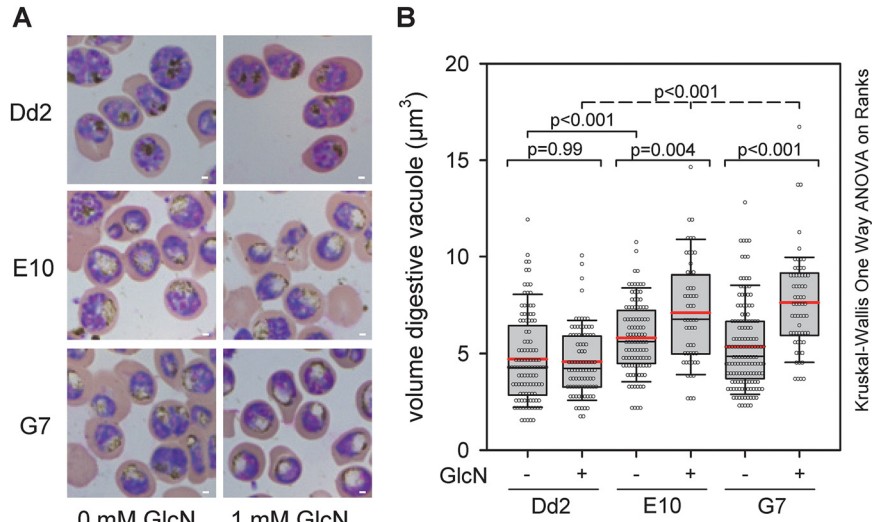

**FIG 5** Altered digestive vacuoles upon induction of the PfCRT knock-down phenotype. (A) Giemsa smear of infected erythrocytes from Dd2 and the PfCRT knock-down mutants E10 and G7 in the presence and absence of glucosamine for 3 days. Note the translucent digestive vacuoles in PfCRT knock-down mutants under induced conditions. Bar, 1 $\mu$m. (B) Size of the digestive vacuole in the parasite lines indicated in the presence and absence of glucosamine (3 days). Data points indicate measurements from at least three independent biological replicates.

delivery to the digestive vacuole (33, 34). As seen in Fig. 6B, there is a clearly visible fluorescence signal in the digestive vacuole in E10 parasites under PfCRT knock-down conditions, as exemplified by pool 1, and not in the absence of GlcN (Fig. 6B). Quantification of the digestive vacuolar fluorescence signals confirmed significant increases in signal intensity across all five pools in response to the PfCRT knock down, as compared with untreated controls (Fig. 6C). Similarly, increased digestive vacuolar fluorescence signals across the five di-peptide pools were observed in Dd2 treated overnight with 0.89 $\mu$M verapamil, a partial mixed type inhibitor of PfCRT (35) (Fig. 6B and C).

We next chose the fluorescently labeled di-peptide Val-Asp (Fig. 6A) for further analysis and again found significant accumulation of fluorescence in the digestive vacuole under PfCRT knock-down conditions in both E10 and G7 and in verapamil treated Dd2, as compared with the untreated controls (Fig. 6D and E). Treatment of the parental strain Dd2 with GlcN did not affect digestive vacuolar fluorescence, which remained at background levels (Fig. 6D and E). The fluorescently labeled di-peptide Val-Asp was found to be stable during the course of the experiment, as shown by metabolite extraction followed by thin-layer chromatography, suggesting that neither human erythrocytes nor the parasite metabolically altered the fluorescently labeled di-peptide (Fig. S1 in the supplemental material).

Comparable data were obtained using the fluorescently labeled single amino acid valine. Again, fluorescence accumulated in the digestive vacuole upon induction of the PfCRT knock-down (Fig. S2 in the supplemental material). However, both the GlcN treated and untreated cells exhibited a growth defect, which might indicate a toxic effect of the fluorescent probe, possibly by inhibiting PfCRT transport activity or interfering with peptide degradation in the digestive vacuole. Fluorescently labeled oligopeptides of 3 (VDP) and 4 (VDPV) residues in length did not accumulate in the digestive vacuole (Fig. S2), most likely because they were not taken up across the erythrocyte membrane, consistent with a preference of the new permeation pathways for lower molecular mass solutes (34).

## DISCUSSION

Previous studies have reported a broad range of partly contradictory putative non-drug related functions of PfCRT, ranging from a role in pH regulation to the transporter of various organic and inorganic ions to PfCRT acting as an oligopeptide carrier (24, 25,

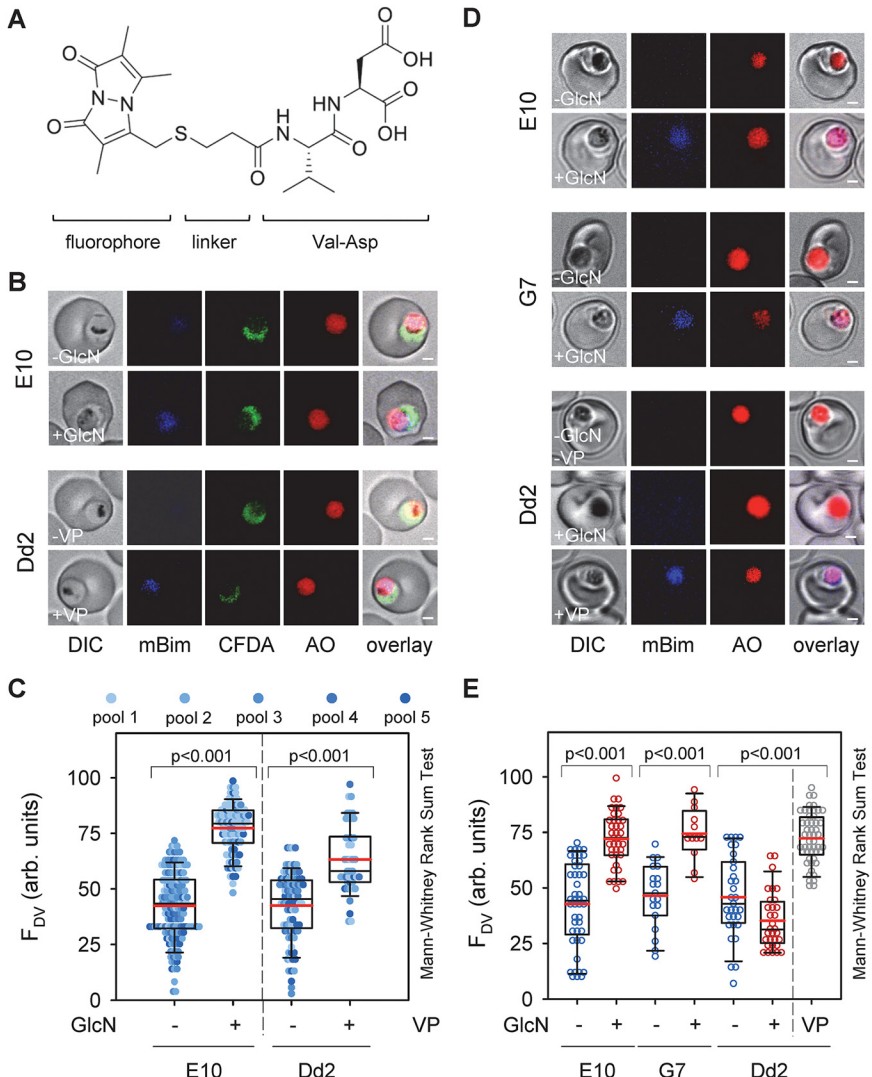

**FIG 6** Accumulation of fluorescently labeled dipeptides in the digestive vacuole of PfCRT knock-down mutants under induced conditions. (A) Chemical structure of a representative fluorescently labeled di-peptide, consisting of the fluorophore monobimane, the linker 3-mercaptopropionic acid, and the amino acids valine and aspartic acid. (B) Representative live cell confocal fluorescence images showing accumulation of fluorescently labeled di-peptides (pool 1; mBim) in the digestive vacuole of the PfCRT knock-down mutant E10 under induced (+GlcN; 1 mM for 3 day) and non-induced (-GlcN) conditions and in Dd2 in the presence (+0.89 $\mu$M VP for 3 day) and absence of verapamil (-VP), a partial mixed type inhibitor of PfCRT$^{Dd2}$ (35). Acridine orange (AO) and 5-carboxyfluorescein diacetate (cFDA) were used as live cell viability probes staining the digestive vacuole and the parasite's cytoplasm, respectively. Bar, 1 $\mu$m. (C) Quantitative analysis of fluorescence signal associated with dipeptides in the digestive vacuole ($F_{DV}$) of the PfCRT knock-down mutant E10 under induced and non-induced conditions and Dd2 in the presence and absence of verapamil. The pools contained the following fluorescently labeled dipeptides: pool 1: KT, AQ, HL, NL, LS; pool 2: GS, LT, PA, SP, GV, VD; pool 3: PK, GH, PV, LP, DK; pool 4: KS, HK, MP, KG, SH, HL; pool 5: QA, LH, SL, TL, GV, SV (Table S9). (D) Like B with the exception that the fluorescently labeled dipeptide VD was investigated. Bar, 1 $\mu$m. (E) Like B with the exception that the fluorescently labeled dipeptide VD was investigated.

27–29). Although all of these proposals have their merits, they were largely based on studies conducted in heterologous expression systems, with little confirmatory evidence in the parasite. Here we report the successful generation of a conditional PfCRT knock-down mutant in *P. falciparum*, together with comparative metabolomics and metabolite-specific fluorescence microscopy. The work has allowed us to shed additional light on the natural function of PfCRT within the parasite system and to reevaluate different proposals in light of our findings.

The *P. falciparum* Dd2 line used in this study encodes a PfCRT variant carrying eight polymorphisms that convert PfCRT into a drug transporting system conferring resistance to the former first line antimalarial drug chloroquine and modulating the responsiveness to related quinoline and quinoline-like antimalarials as well as to structurally unrelated compounds (9, 11–13). The selection of Dd2 as the parental strain for the conditional PfCRT mutation allowed us to verify the knock-down strategy by determining the responses to various drugs in the presence and absence of GlcN. Our finding that inducing the knock-down rendered the Dd2 mutants susceptible to chloroquine and desethyl amodiaquine (Fig. 2) and, in addition, incurred a fitness cost (Fig. 1F), is consistent with PfCRT playing a pivotal role in drug responses and parasite viability (11, 19, 36). Moreover, this finding validates the knock-down strategy and demonstrates that the amount of PfCRT is sufficiently reduced under knock-down conditions to observe defined PfCRT-linked phenotypes.

Semi-quantitative Western analyses estimates the reduction in PfCRT levels to be ∼75% (Fig. 1C and D). The remaining PfCRT activity may explain why the $IC_{50}$-values and accumulation ratios of chloroquine and desoxy amodiaquine approach those of the drug sensitive *P. falciparum* line HB3, but do not reach them (Fig. 2). Residual PfCRT may also be responsible for the unaltered responsiveness to quinine and quinidine under restrictive versus permissive conditions. In this context, it is worth mentioning that quinine and quinidine responsiveness are multifactorial phenotypes and that PfCRT is only one of several contributing factors (37–40).

The combined unbiased and targeted metabolomics approaches conducted herein revealed a significant accumulation of oligopeptides in the knock-down mutants upon induction (Fig. 3). While this result would link PfCRT to altered oligopeptide handling, the metabolomics studies did not provide spatial information regarding the subcellular compartment in which the oligopeptides accumulated, although the digestive vacuole would be a plausible site. Such spatial information could be retrieved from studies conducted with fluorescently labeled di-peptides (Fig. 6), which identified the digestive vacuole as the subcellular compartment in which the oligopeptides accumulated.

The accumulating oligopeptides were structurally diverse in terms of amino acid composition and length, varying from 2 to 8 amino acids, with di-, tri, and tetrapeptides being the dominant species (Fig. 4B and Tables S3 to S8 in the supplemental material). We could not test for oligopeptides longer than 8 residues since our mass spectrometry method had an upper molecular mass detection limit of ∼1200 Da. Despite their overall diversity, most of the oligopeptides had an isoelectric point between 4.0 and 5.4 (Fig. 4B) and were electrically neutral or carried a single positive or negative charge at the digestive vacuolar pH of 5.2 (Fig. 4B). Almost all of the oligopeptides could be related to one or other of a group of 12 of the most abundant erythrocyte proteins, suggesting that they originated from proteolytic processes occurring in the digestive vacuole (Tables S3 to S8). The observed decrease in the amount and diversity of oligopeptides with time of knock-down induction (Fig. 4B) might be due to a feed-back mechanism inhibiting further hemoglobin uptake and/or degradation. The findings that the parasites were able to enzymatically convert CFDA into its fluorescent form 3 days after the addition of GlcN to induce the PfCRT knock-down (Fig. 6B and D), suggests that the parasites still maintained viability.

On the basis of our findings we propose that PfCRT is an oligopeptide carrier in the parasite, with a broad specificity for neutral and mono-charged oligopeptides with a length of two to at least eight residues. Our finding that fluorescently labeled dipeptide pools and the mBim-conjugated Val-Asp, as well as mBim-conjugated valine, accumulated in the digestive vacuole upon PfCRT reduction provides further support for this model. However, with the fluorophore added these sensors are enlarged and may resemble tripeptides and dipeptides regarding substrate identity. In the case of the fluorescently labeled amino acid valine, we noted a fitness cost (Fig. S1 in the supplemental material), consistent with previous studies showing a toxic and lytic activity of some dipeptide derivatives (41). Our conclusions are consistent with a recent study showing

that PfCRT can transport oligopeptides 3 to 11 amino acids in length under trans-stimulation conditions in the heterologous *X. laevis* oocyte system (29).

We acknowledge the limitation of our study in the sense that most findings are based on studies conducted with a *P. falciparum* strain that carries a variant PfCRT protein. Efforts to generate a conditional PfCRT knock-down in a wild-type strain have thus far failed in our laboratory in spite of numerous attempts. Although it is undisputed that mutational changes can alter the substrate specificity of PfCRT, as has clearly been demonstrated for antimalarial drugs (12, 13), a variant-transcending role of PfCRT as an oligopeptide carrier is consistent with recent *in vitro* studies conducted in *X. laevis* oocytes, showing oligopeptide transport by both wild type and variant forms of PfCRT, including PfCRT[Dd2], albeit drug resistance conferring variants displayed a reduced transport capacity in terms of overall activity and oligopeptide diversity (29).

It is further possible that the genetic alterations introduced in the *pfcrt* locus affected the transport efficiency of PfCRT. In this context, a previous study has shown that altering the 3′ untranslated region of *pfcrt* decreased protein expression levels and tolerance to chloroquine (42). Although we observed comparable PfCRT protein levels in Dd2 and the PfCRT mutants in the absence of GlcN (Fig. 1C and D), we cannot exclude an effect of the C-terminally fused hemagglutinin tag on the transport activity of PfCRT. Such an effect might explain the already slightly enlarged digestive vacuoles in the mutants in the absence of GlcN (Fig. 5B). However, this putative limitation is not seen as an obstacle to the interpretation of our results, or the conclusions drawn from them.

A function of PfCRT in oligopeptide transport across the digestive vacuolar membrane would explain the essential nature of PfCRT for intraerythrocytic parasite development. To meet its nutrient requirements and prevent the host cell from lysis due to increased colloid osmotic pressure generated by the growth of the parasite and its anabolic activities, the parasite gradually degrades hemoglobin and other erythrocyte proteins in its digestive vacuole (2, 43). The resulting oligopeptides would be transported by PfCRT into the cytoplasm, where they are degraded to single amino acids by neutral aminopeptidases (44). The amino acids are then recycled for biosynthesis purposes and/or expelled to maintain the colloid osmotic balance. On the other hand, accumulation of the osmotically active oligopeptides in the digestive vacuole would cause swelling and eventually rupture of this organelle.

## MATERIALS AND METHODS

**Radio-chemicals.** Radiolabeled antimalarial drugs were obtained from the following vendors: [³H]-chloroquine (18.8 Ci/mmol), GE Healthcare; [³H]-quinine (20 Ci/mmol), American Radiolabeled Chemicals, Inc; and [³H]-amodiaquine (20 Ci/mmol), Moravek Biochemicals, Inc.

**Primers.** All oligonucleotide primers used in this study are listed in Table S10 in the supplemental material.

**Parasite culture.** The *P. falciparum* line Dd2 and the PfCRT knock-down mutants were cultured as described (45), using RPMI 1640 medium supplemented with 0.5% (w/v) Albumax II (ThermoFisher Scientific), 200 $\mu$M hypoxanthine, 20 $\mu$g mL$^{-1}$ gentamicin and 5% human serum. The cultures were maintained at a 4% hematocrit, using fresh HbAA erythrocytes (within 10 days after blood donation), 5% parasitemia, 37°C and 5% $O_2$, 3% $CO_2$ and 96% humidity. Cultures were repeatedly synchronized, using 5% sorbitol (46). Throughout the study trophozoites (22- to 28-h post invasion) were used.

**Generation of PfCRT knock-down mutants.** To generate the PfCRT knock-down mutant, a fragment of ~0.7 kb was amplified by PCR from genomic DNA of the *P. falciparum* line Dd2, using primers Pfcrt-1-IF-for and Pfcrt-Bgl II-rev. A second fragment of ~0.9 kb containing the remaining part of the *pfcrt* gene was amplified from a plasmid harboring a re-codonized Dd2 *pfcrt* gene (GeneArt, ThermoFisher Scientific), using the primers Pfcrt[rc]-Bgl II-for and Pfcrt[rc]-1280-IF-no stop-rev. Both fragments were digested with the restriction endonuclease Bgl II before the fragments were ligated. The ligated product was amplified using primers Pfcrt-1-IF-for and Pfcrt[rc]-1280-IF-no stop-rev. The final PCR fragment was cloned into the SpeI/BssHII sites of pL6-HA-gmlS (47) by In Phusion (TaKaRa). In addition, a 0.3 kb fragment containing the 3′ UTR of *pfcrt* amplified from genomic DNA, using primers Pfcrt-3'UTR-IF-for and Pfcrt-3'URT-AflII-rev, was cloned into the Nar I and Afl II restriction sites of the pL6-HA-gmlS vector, yielding pL6-pfcr-HA-glmS-3′UTR. The guide RNA (Table S10 in the supplemental material) was cloned into the BtgZ1 site of the pL6-pfcr-HA-glmS-3′UTR. The resulting transfection vector was verified by sequencing analysis. Ring stages of the *P. falciparum* line Dd2 were transfected by electroporation with 75 $\mu$g each of the pL6-*pfcrt*-HA-gmlS-3′UTR and the pUF1-Cas9 transfection vector (30). Transfectants were selected using 5 nM WR99210 and 1.5 $\mu$M DSM1. Clonal lines were obtained by limiting dilution and the genetically modified *pfcrt* locus was confirmed by amplification and sequencing reaction (see Table S10 for primers).

**Treatment with glucosamine.** Highly synchronized cultures at the early ring stage were treated with 1 mM glucosamine for 2 days before cells were synchronized once more, and treatment with glucosamine was continued for an additional day until parasites developed to trophozoites (22 to 28 h post invasion). At this point cells were harvested for Western analysis, drug accumulation assays, and 3 days metabolomics analyses. For the 1-day metabolomics study, cells were harvested 24 h after addition of glucosamine.

**Asexual intraerythrocytic proliferation.** Highly synchronized parasite cultures at the ring stage (0.5% parasitemia) were split in two aliquots and one aliquot was incubated in the presence of 1 mM GlcN for 72 h. The other aliquots served as the untreated controls. Parasite proliferation was assessed using Giemsa-stained thin blood smears. Pyknic and crisis forms, as occurred in GlcN treated PfCRT knock-down parasite cultures, were not considered. The cumulative parasitemia was calculated as the sum of the parasitemia in the culture after considering splitting and dilution factors.

**Fitness assay.** The relative fitness of the PfCRT knock-down mutants was determined as described (19). Briefly, mixed cultures of Dd2 and the PfCRT knock-down mutants were cultured in the presence or absence of 1 mM GlcN. The allelic proportions were measured by pyrosequencing during 20 cycles, using the primers PfCRT-Pyro-for-biotin and PfCRT-Pyro-rev. The sequencing reaction was carried out using primer PfCRT-Pyro-seq. The proportion of Dd2 in the cultures is represented on the *y* axis as a function of time. The data represent the mean $\pm$ SEM of three independent determinations using blood from different donors. Note that the results of the fitness assay lags those of the proliferation assay, particularly during the first days following treatment with GlcN, owing to the different experimental designs of the two assays and their different read-out parameters: presence of a DNA fragment, following PCR amplification, independent of whether the DNA comes from healthy or dying parasite versus number of healthy parasites according to microscopic examination.

**Drug response assays.** Cell proliferation assays to determine drug $IC_{50}$-values were performed as described (48), using the SYBR green fluorescence-based assay (49). Briefly, highly synchronized parasite cultures at the ring stage (0.5% parasitemia, 3% hematocrit) were incubated in the presence and absence of glucosamine (1 mM) and increasing concentration of the drugs indicated. Cells were harvested 72 h later and were lysed in lysis buffer (20 mM Tris-HCl, 5 mM EDTA, 0.008% saponin, 0.08% Triton X-100, pH 7.4) containing SYBR green. Fluorescence was determined using a FLUOstar OPTIMA plate reader with the following settings: excitation wavelength: 485 nm; emission wavelength: 520 nm; gain: 1380; number of flashes per well: 10; top optic.

The drug accumulation assays were performed as described (48, 50). Briefly, magnet purified *P. falciparum*-infected erythrocytes at the trophozoite stage (22 to 28 h post invasion) were resuspended in 200 $\mu$L prewarmed reaction buffer A (bicarbonate-free RPMI 1640) containing 11 mM glucose, 25 mM HEPES, 2 mM glutamine and 40 nM the radiolabeled drugs indicated (pH 7.3) at an hematocrit of 25000 cells $\mu L^{-1}$, as determined using an automated cell counter (Z1-Coulter Particle Counter, Beckman Coulter Inc.). Cells were then incubated for 20 min at 37°C before duplicative 75 $\mu$L aliquots were removed and diluted with an equal volume of ice-cold reaction buffer A. The aliquots were immediately spun through a layer of a 5:4 mixture of dibutyl phthalate and dioctyl phthalate (15,000 $\times$ *g*, 5 s). The aqueous phase containing the unincorporated radiolabeled drug, was removed and transferred to a scintillation vial to determine the extracellular drug concentration for each reaction. The cell pellets were recovered by cutting the reaction tubes through the oil layer with a sharp scalpel while squeezing the tubes with tweezers. The tips of the tubes containing the cell pellets were placed in 1.5 mL Eppendorf tubes and incubated with 66 $\mu$L ethanol and 33 $\mu$L tissue solubilizer overnight at 55°C. The lysates were decolorized by the addition of 25 $\mu$L of 30% $H_2O_2$ and acidified by the addition of 25 $\mu$L of 1 N HCl. The lysates were transferred to scintillation vials, and the radioactivity was measured using a liquid scintillation counter (TRI-CARB 2100 TR, Packard). The intracellular drug concentration was calculated from the amount of radiolabeled drug taken up by the cells and by assuming a volume of 75 fL for a trophozoite-infected erythrocyte (51). Drug accumulation was expressed as the ratio of the intracellular versus the extracellular drug concentration ($Drug_{in}/Drug_{out}$).

**Western analysis.** Cells were treated for 3 days with 1 mM glucosamine, as described above, before trophozoite stages were enriched by magnet purification and lysed in PBS containing 0.07% saponin and protease inhibitors (1 mM phenylmethylsulfonyl fluoride, 50 $\mu$g mL$^{-1}$ aprotinin, and 20 $\mu$g mL$^{-1}$ leupeptin). The liberated parasites were collected by centrifugation. The parasite pellet was then washed two times in PBS and resuspended in protein loading buffer (250 mm Tris, pH 6.8, 3% SDS, 20% glycerol, 0.1% bromophenol blue) and sonicated briefly. A supernatant fraction was collected after centrifugation (17,000 $\times$ *g* for 30 min at 4°C) and subsequently analyzed by SDS-PAGE. Gels were transferred to a polyvinylidene difluoride membrane (Bio-Rad) using an iBlot2 transfer apparatus. The following antibodies were used: *i*) primary antibodies, mouse anti-$\alpha$-tubulin (1:1000 dilution; clone B-5-1-2; Sigma-Aldrich) and guinea pig anti-PfCRT (1:1000 dilution) (26); and *ii*) secondary antibodies, donkey anti-mouse POD (1:10,000 dilution; Abcam), and goat anti-guinea pig POD (1:10,000 dilution; Abcam). All antibodies were diluted in PBS containing 1% (wt/vol) BSA. The signal was captured with a blot scanner (C-DiGit from LI-COR Biosciences).

**Metabolomics.** Cultures of the parental *P. falciparum* line Dd2 and the two PfCRT knock-down mutants E10 and G7 were harvested at $\sim$ 5% parasitemia at the trophozoite stage after treatment for 1 or 3 days with 1 mM glucosamine. Cells were magnet purified and washed with PBS. 3 $\times$ $10^7$ cells were then extracted in 150 $\mu$L extraction buffer (chloroform/methanol/water 1:3:1 vol/vol/vol) and incubated at 4°C for 15–30 min. Extracts were centrifuged at 17.000 $\times$ *g* for 10 min at 4°C and 100 $\mu$L of the particle free supernatant was transferred to a screw cap micro-centrifuge tube. Samples were stored at −80°C until analysis. 4 and 3 independent biological replicates were analyzed for the 1- and 3-day induction experiments, respectively. Biological replicates differed from one another by time and blood donor. Mass spectrometry was performed as previously described (52, 53). Briefly, samples were analyzed on a

Thermo Scientific Q-Exactive Orbitrap mass spectrometer running in positive/negative switching mode. This was connected to a Dionex UltiMate 3000 RSLC system (Thermo Fisher Scientific) using a ZIC-pHILIC column (150 mm × 4.6 mm, 5 $\mu$m column, Merck Sequant). The column was maintained at 30°C and samples were eluted with a linear gradient (20 mM ammonium carbonate in water, solution A and acetonitrile, solution B) over 26 min at a flow rate of 0.3 mL min$^{-1}$ as follows: 0 to 20 min 20%-to 80% solution A, 15 to 17 min 95% solution A, 17 to 26 min 20% solution A. The injection volume was 10 $\mu$L and samples were maintained at 5°C prior to injection. Mass spectrometry data was processed using a combination of XCMS 3.2.0 (54) and MZMatch.R 1.0–4 (55). Briefly, data were converted from Thermo proprietary raw files to the open format mzXML. Unique signals were extracted using the centwave algorithm and matched across biological replicates based on mass to charge ratio and retention time. These grouped peaks were then filtered based on relative standard deviation and combined into a single file. The combined sets were then filtered on signal to noise score, minimum intensity and minimum detections. The final peak set was then gap-filled and converted to text for use with IDEOM v18 (56).

Putative metabolite identification corresponds for the most part to Metabolite Standards Initiative (MSI) level 2 (mass only), whereas metabolites matching in retention time to an included standard correspond to level 1, as indicated in the underlying data. Peaks having an area with root squared deviation across pooled samples > 50% were excluded, as were those with a retention time < 4 min (due to poor resolution). Peptides were filters for masses between 70 and 1,000 *m/z*, with charges from 1 to 3 for each peptide. Unfortunately, the IDEOM analysis software does not correctly import peptides of higher molecular mass. Oligopeptides of higher molecular mass were, therefore, identified in a targeted approach by matching the masses from the raw metabolomics database to putative peptides from the 12 most abundant red blood cell proteins. No additional filtering was done to remove redundant or duplicate peptides, nor the charge in the mass spectrum verified for every peak as this would be very time-consuming to do manually for every unique mass. Comparative analysis was, as described (57), using a Pearson correlation, with the first variable being the signal intensities of the identified metabolites and the second variable being 1 and 0 for the glucosamine treatment group (consisting of each of the four or three independent biological replicates of E10 and G7) and the non-treated control group (consisting of each of the four or three independent biological replicates of E10, G7, and Dd2 in the presence and absence of glucosamine), respectively. The resulting Pearson coefficient was converted into a *P*-value, from which the LOD score was calculated. The fold change for each metabolite was calculated by dividing the mean signal intensity obtained for each treated PfCRT knock-down mutant by the mean signal intensity of the corresponding untreated control.

**Live cell imaging.** Dd2 and the PfCRT knock-down mutants were cultured in the presence and absence of 1 mM glucosamine for 3 days. In parallel Dd2 was cultured in the presence of 0.89 $\mu$M verapamil for 3 days. Highly synchronized parasite cultures were used. At day 2 of treatment with or without glucosamine or verapamil (ring stages), cells were transferred to microtiter plates and fluorescently labeled di-peptides (final concentration of 2.5 mM, termed mBim in the fig. 6) were added to the cultures. The di-peptides were coupled to bromobimane via a 3-mercaptopropionic acid linker (Pepscan). The fluorescently labeled oligopeptides used in this study are listed in Table S9 in the supplemental material. The single oligopeptides investigated were inspired by the recent study showing that VDPVNF cis-inhibited chloroquine transport by PfCRT$^{Dd2}$ in the *X. laevis* oocyte system (29) 24 h later, cells were collected and washed twice with 37°C prewarmed Ringer's solution (122.5 mM NaCl, 5.4 mM KCl, 1.2 mM CaCl$_2$, 0.8 MgCl$_2$, 11 mM glucose, 10 mM HEPES, 1 mM NaH$_2$PO$_4$, pH 7.4) before cells were incubated in Ringer's solution containing 5 $\mu$M acridine orange (AO, Sigma-Aldrich) and/or 10 $\mu$M 6-chloromethyl-2, 7-dichlorodihydrofluorescein diacetate (CFDA, ThermoFisher Scientific). Cells were washed and visualized by confocal laser scanning microscopy (Zeiss LSM510) equipped with UV and visible laser lines and a C-Apochromat objective with a 63× magnification. Fluorescence signals were collected in Multi-Track mode, using the following laser lines and filter settings: For fluorescently labeled di-peptides: excitation wavelength 364 nm; emission detected using a LP 385-nm filter; for AO and CFDA: excitation wavelength 488 nm; emission detected using an LP 650 nm filter (for AO) and an LP 505–550 nm band pass filter (for cFDA). The images were acquired using the LSM imaging software and processed with the Fiji program.

**Morphological measurement.** Images of infected erythrocytes stained with acridine orange (see above) were analyzed using Fiji and the area of the digestive vacuole was quantified. Assuming a spherical shape of the digestive vacuole (58), its volume was calculated based on the following equation: V = 4/3 $\pi r^3$, where r is the radius.

**Calculation of isoelectric point and charge.** Physicochemical properties of the oligopeptides were calculated using the web-based isoelectric point calculator 2 (59).

**Thin layer chromatography.** Samples were prepared as described under life cell imaging. Cells were washed several times with PBS at 4°C before the cells were extracted with chloroform/methanol/water (1:3:1 vol/vol/vol). Extracts were subsequently dried, using a SpeedVac, and resuspended in 10 $\mu$L of chloroform/methanol/water mixture. Samples were loaded on an activated 0.2 mm silica gel TLC plate (N-HR/UV254, Macherey-Nagel). The chromatography was run using a methanol:water mixture (7:3, vol/vol) containing 0.1 M ammonium acetate (pH =7.0). Fluorescence signals were visualized using an Amersham Imager 600.

**Statistical analysis.** Where indicated, data points were overlaid by a box plot, with median (black line), mean (red line) and the 25% and 75% quartile range being shown. The error bars indicate the 90th and the 10th percentile. Statistical tests used for data analysis are referenced in the figures. $P < 0.01$ was considered significant. Data were analyzed using Sigma Plot version 13 (Systat Software).

**Data availability.** The data that support the findings of this study, including the supplementary Tables, are openly available at DRYAD: https://doi.org/10.5061/dryad.573n5tb9c.

## SUPPLEMENTAL MATERIAL

Supplemental material is available online only.

**SUPPLEMENTAL FILE 1**, PDF file, 0.1 MB.

## ACKNOWLEDGMENTS

We thank M. Müller and A. Kernaja for excellent technical assistance. M.P.B. was funded by a Wellcome Trust core grant to the Wellcome Centre for Integrative Parasitology (104111/Z/14/Z).

We declare no conflict of interest.

C.P.S., M.P.B., and M.L. conceived and designed the experiments. C.P.S., E.M., S.M.C., L.M., and S.W. performed the experiments. C.P.S., M.P.B., and M.L. wrote the manuscript. All authors participated in data analysis and interpretation. All authors have read and edited the manuscript and agree with the final version.

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
