## [Reviewer comments · Microbiology Spectrum]

Microbiology Spectrum

The knock-down of the chloroquine resistance transporter PfCRT is linked to oligopeptide handling in *Plasmodium falciparum*

Cecilia Sanchez, Erin Manson, Sonia Moliner-Cubel, Luis Mandel, Stefan Weidt, Michael Barrett, and Michael Lanzer

Corresponding Author(s): Michael Lanzer, Heidelberg University

Review Timeline:

Submission Date:	March 26, 2022
Editorial Decision:	April 20, 2022
Revision Received:	June 9, 2022
Accepted:	June 27, 2022

Editor: Anat Florentin

Reviewer(s): Disclosure of reviewer identity is with reference to reviewer comments included in decision letter(s). The following individuals involved in review of your submission have agreed to reveal their identity: Benjamin Liffner (Reviewer #1)

Transaction Report:

DOI: <https://doi.org/10.1128/spectrum.01101-22>

April 20, 2022

Prof. Michael Lanzer
Heidelberg University
Center of Infectious Diseases, Parasitology
Heidelberg D-69120
Germany

Re: Spectrum01101-22 (The knock-down of the chloroquine resistance transporter PfCRT is linked to oligopeptide handling in *Plasmodium falciparum*)

Dear Prof. Michael Lanzer:

Link Not Available

Sincerely,

Anat Florentin

Journals Department
Reviewer comments:

Reviewer #1 (Comments for the Author):

Summary

The chloroquine resistance transporter of *Plasmodium falciparum* (PfCRT) has long been of significance to malaria parasite biologists for the role it plays in conferring resistance to the former frontline antimalarial chloroquine. Despite this, relatively little is known about the function of this transporter outside of its role in chloroquine transport. Moreover, most studies regarding CRT have used heterologous systems, making it difficult to extrapolate its in situ function(s) in malaria parasites. The authors utilise the GlmS ribozyme knockdown system to show that CRT is important for parasite growth and long-term fitness. Additionally, the authors show that CRT knockdown parasites become have increased chloroquine sensitivity and accumulation. Using both

targeted and untargeted metabolomics, that authors show that the majority of the metabolites that are differentially detected when CRT is knocked down are small peptides, predominantly 2-4 amino acids long. Using microscopy, the authors then show that CRT is involved in the transport of small peptides out of the digestive vacuole, with CRT knockdown parasites exhibiting bloated digestive vacuoles and an accumulation of small peptides. Collectively, the authors robustly show that one of the functions of CRT in *P. falciparum* is the transport of small peptides. The conclusions drawn in this study are well supported by the data, and the authors should be commended for their attention to detail in regards to statistical analysis and robustness. I have some comments, principally regarding the amount of detail provided in the methods sections about the experiments in this study, but I suspect these will be relatively easily addressed by the authors.

Major comments:

- While the application of the GlmS knockdown system to Dd2 parental parasites has allowed the authors to study the function of CRT knockdown in a chloroquine-resistant parasite, the authors also note that the 8 mutations present in Dd2 CRT alter its activity. Considering this, the authors should mention in the discussion that it is currently unclear how similar the wildtype-functions of CRT are in chloroquine-sensitive and chloroquine-resistant parasites.
- Compared to the parental Dd2 parasites, E10 and G7 showed enlarged vacuoles and decreased fitness, even in the absence of glucosamine treatment. Because this is also the observed phenotype of CRT knockdown, this should be mentioned by the authors in the discussion, and if this is a phenomenon that has previously been observed using the GlmS system, the authors should discuss this too.
- In the methods, the authors should explain how the size of the digestive vacuole was measured. Additionally, the authors should provide details on how the fluorescence of the digestive vacuole was measured and how this was controlled.
- Further details should be provided on how the parasite growth assays were performed, including whether the parasites were synchronised and at what stage of the lifecycle glucosamine was added, currently it is unclear whether there are two rounds of invasion or 1 in Figure 1E.
- Further details should be included about the parasite cultures used in Figure 2. Based on the details provided, it is currently unclear if, or how, the level of parasite growth inhibition caused by PfCRT knockdown could influence CQ and dAQ IC50 values.
- In Figure 6, the authors use a good control of VP to show that inhibition of CRT can lead to dipeptide accumulation, however, it seems there was no glucosamine control performed on Dd2 parasites. To rule out a possible influence of glucosamine on dipeptide accumulation, the authors should also include a Dd2 + GlcN control.

Minor comments:

- Missing italics of species, gene names, or Latin in lines 58, 60, 100, 154.
- The authors often refer to the generated mutants as knockdown mutants, and the presence of glucosamine as the induced state. I found this a little confusing, particularly in the first two results paragraphs. It may be more clear to simply refer to them as in the presence or absence of glucosamine as this terminology can be applied to both Dd2 and the generated mutants.
- Lines 118-120 refer to Dd2 in Figure 3, but it seems that only results from E10 and G7 are included in Figure 3.
- It seems that the number of metabolites in Figure 4A and peptides in 4B are pooled from experiments of both E10 and G7. If this is the case, the authors should clarify this, or alternatively clarify if the results are only from a single clone.
- Line 180 says parasites were treated with 0.8 mM verapamil, but elsewhere in the text says 0.89 μ m.
- Is there any particular reason the authors chose Val-Asp as the specific dipeptide? If so, this choice should be justified in the text.
- The final paragraph of the results section does not include a reference to Supplementary Figure 1.
- In the Methods, the subtitle that reads "Parasite culture and transfection" only mentions parasite culture, and transfection is covered in the next section.
- The machine used to quantify SYBR green fluorescence should be listed in the methods section.
- The authors should provide further details on how the drug accumulation assay works, and list the machine used to measure the radiation from the label.
- An Anti-POD antibody is listed in the Western blot section of the methods, but this antibody does not appear to have been used in this study.
- The legend of Figure 2 does not contain sufficient information to fully understand the figure. The authors should briefly explain how parasite growth and drug accumulation was measured for these assays.
- From figure 6, it isn't clear what the function of verapamil is, this should be briefly listed in the legend.
- In Figure 1A, there are a series of black lines along PfCRT that don't seem to be explained in the figure legend.
- In the fitness assay, none of the sequence proportions seem to drop significantly until 5 replicative cycles (~10 days). Yet in the growth assays, the glucosamine treated G7 and E10 parasites have only ~1/3 the parasitaemia of untreated after only 3 days. Can the authors provide any insight or justification on this discrepancy?
- The authors state in the figure legend that all datapoints in Figure 2 represent biological replicates. Given the number of datapoints, and how concordant many of the values are (like dAQ in A, for example), I wonder if these graphs actually include technical replicate values as well. Can the authors please clarify if this is the case.
- In Figure 2B, Drugin/Drugout referring to intracellular and extracellular accumulation should be specified in the figure legend.
- The range of values included in each column of the histograms in Figure 4B are unclear, especially for the isoelectric point. Perhaps the graph could be expanded to include more columns, or the ranges specified along the x-axis.

This is an elegant, simple and well-presented work to assess the natural biological role of PfCRT. The data is clearly presented and supports very well the new proposed function of PfCRT in oligopeptide transport. The rigor of the results and the experimental design is high.

Two minor suggestions:

- 1) Figure S1 does not really show a fitness cost. May be adding a growth curve or giemsa stained pictures will aid the readers in interpreting the results and conclusions from the authors.
- 2) 5% albumax seems quite high as it is usually used at 0.5% wt/vol (or 5 g/L). please clarify the units.

Reviewer #3 (Comments for the Author):

Title: The knock-down of the chloroquine resistance transporter PfCRT is linked to oligopeptide handling in *Plasmodium falciparum*

The study by Sanchez et al. shows that chloroquine resistance transporter PfCRT is involved in the transportation of the oligopeptides from the digestive vacuole of the parasite to the cytoplasm. These peptides are believed to be cleaved to single amino acids and are then recycled for biosynthesis purposes and/or expelled to maintain the colloid osmotic balance. This observation is in agreement with a recent study (Shafik et al., 2020, Nature Communications) as indicated by the authors. The quality of the data generated in this study is very good and it is well supported with statistics as and when required.

However, I have a few major concerns in this study which are listed below:

1. The major premise of this work is that the non-drug related function of the PfCRT is less clear and the authors wanted to dissect the "Natural Function" of PfCRT. Also, as mentioned by the authors and known in the literature that several mutations acquired in the PfCRT affect its transporting property. This is also evident from the recent study by Shafik et al., 2020, Nature Communications where the transporting properties of PfCRT in chloroquine-sensitive and resistant strains are distinct.

In this study, the authors have used the Dd2 strain, which is a chloroquine-resistant strain with eight polymorphisms that convert PfCRT to a transporter system that confers drug resistance. Thus, the effect observed in this study, are at best in comparison to the mutant PfCRT (mutations from the resistant strain). Therefore, to assign a physiological function to PfCRT, the authors should do these assays in a chloroquine-sensitive strain.

2. Figure 1 C and D: Even though a decrease is observed upon induction with glucosamine, there seems to be a significant increase in PfCRT expression in the uninduced knockdown parasite lines (E10 and G7) as compared to the Dd2 parent line. It is not clear if the tagging of the endogenous locus leads to this effect and in turn may have an effect on the transporting property of PfCRT. The authors need to clarify this effect.

3. Authors can also include a brief explanation on the accumulation and localization of specifically dipeptides in the digestive vacuole in Dd2 and conditional knockdown lines with respect to chloroquine resistance. Do authors expect this to be the same in a chloroquine-sensitive line?

Staff Comments:

Preparing Revision Guidelines

For complete guidelines on revision requirements, please see the journal Submission and Review Process requirements at <https://journals.asm.org/journal/Spectrum/submission-review-process>. **Submissions of a paper that does not conform to**

Microbiology Spectrum guidelines will delay acceptance of your manuscript. "

Please return the manuscript within 60 days; if you cannot complete the modification within this time period, please contact me. If you do not wish to modify the manuscript and prefer to submit it to another journal, please notify me of your decision immediately so that the manuscript may be formally withdrawn from consideration by Microbiology Spectrum.

Summary

The chloroquine resistance transporter of *Plasmodium falciparum* (PfCRT) has long been of significance to malaria parasite biologists for the role it plays in conferring resistance to the former frontline antimalarial chloroquine. Despite this, relatively little is known about the function of this transporter outside of its role in chloroquine transport. Moreover, most studies regarding CRT have used heterologous systems, making it difficult to extrapolate its *in situ* function(s) in malaria parasites. The authors utilise the GlmS ribozyme knockdown system to show that CRT is important for parasite growth and long-term fitness. Additionally, the authors show that CRT knockdown parasites become have increased chloroquine sensitivity and accumulation. Using both targeted and untargeted metabolomics, that authors show that the majority of the metabolites that are differentially detected when CRT is knocked down are small peptides, predominantly 2-4 amino acids long. Using microscopy, the authors then show that CRT is involved in the transport of small peptides out of the digestive vacuole, with CRT knockdown parasites exhibiting bloated digestive vacuoles and an accumulation of small peptides. Collectively, the authors robustly show that one of the functions of CRT in *P. falciparum* is the transport of small peptides. The conclusions drawn in this study are well supported by the data, and the authors should be commended for their attention to detail in-regards to statistical analysis and robustness. I have some comments, principally regarding the amount of detail provided in the methods sections about the experiments in this study, but I suspect these will be relatively easily addressed by the authors.

Major comments:

- While the application of the GlmS knockdown system to Dd2 parental parasites has allowed the authors to study the function of CRT knockdown in a chloroquine-resistant parasite, the authors also note that the 8 mutations present in Dd2 CRT alter its activity. Considering this, the authors should mention in the discussion that it is currently unclear how similar the wildtype-functions of CRT are in chloroquine-sensitive and chloroquine-resistant parasites.
- Compared to the parental Dd2 parasites, E10 and G7 showed enlarged vacuoles and decreased fitness, even in the absence of glucosamine treatment. Because this is also the observed phenotype of CRT knockdown, this should be mentioned by the authors in the discussion, and if this is a phenomenon that has previously been observed using the GlmS system, the authors should discuss this too.
- In the methods, the authors should explain how the size of the digestive vacuole was measured. Additionally, the authors should provide details on how the fluorescence of the digestive vacuole was measured and how this was controlled.
- Further details should be provided on how the parasite growth assays were performed, including whether the parasites were synchronised and at what stage of the lifecycle glucosamine was added, currently it is unclear whether there are two rounds of invasion or 1 in Figure 1E.
- Further details should be included about the parasite cultures used in Figure 2. Based on the details provided, it is currently unclear if, or how, the level of parasite growth inhibition caused by PfCRT knockdown could influence CQ and dAQ IC50 values.

- In Figure 6, the authors use a good control of VP to show that inhibition of CRT can lead to dipeptide accumulation, however, it seems there was no glucosamine control performed on Dd2 parasites. To rule out a possible influence of glucosamine on dipeptide accumulation, the authors should also include a Dd2 + GlcN control.

Minor comments:

- Missing italics of species, gene names, or Latin in lines 58, 60, 100, 154.
- The authors often refer to the generated mutants as knockdown mutants, and the presence of glucosamine as the induced state. I found this a little confusing, particularly in the first two results paragraphs. It may be more clear to simply refer to them as in the presence or absence of glucosamine as this terminology can be applied to both Dd2 and the generated mutants.
- Lines 118-120 refer to Dd2 in Figure 3, but it seems that only results from E10 and G7 are included in Figure 3.
- It seems that the number of metabolites in Figure 4A and peptides in 4B are pooled from experiments of both E10 and G7. If this is the case, the authors should clarify this, or alternatively clarify if the results are only from a single clone.
- Line 180 says parasites were treated with 0.8 mm verapamil, but elsewhere in the text says 0.89 μm .
- Is there any particular reason the authors chose Val-Asp as the specific dipeptide? If so, this choice should be justified in the text.
- The final paragraph of the results section does not include a reference to Supplementary Figure 1.
- In the Methods, the subtitle that reads “Parasite culture and transfection” only mentions parasite culture, and transfection is covered in the next section.
- The machine used to quantify SYBR green fluorescence should be listed in the methods section.
- The authors should provide further details on how the drug accumulation assay works, and list the machine used to measure the radiation from the label.
- An Anti-POD antibody is listed in the Western blot section of the methods, but this antibody does not appear to have been used in this study.
- The legend of Figure 2 does not contain sufficient information to fully understand the figure. The authors should briefly explain how parasite growth and drug accumulation was measured for these assays.
- From figure 6, it isn't clear what the function of verapamil is, this should be briefly listed in the legend.
- In Figure 1A, there are a series of black lines along PfCRT that don't seem to be explained in the figure legend.
- In the fitness assay, none of the sequence proportions seem to drop significantly until 5 replicative cycles (~10 days). Yet in the growth assays, the glucosamine treated G7 and E10 parasites have only ~1/3 the parasitaemia of untreated after only 3 days. Can the authors provide any insight or justification on this discrepancy?

- The authors state in the figure legend that all datapoints in Figure 2 represent biological replicates. Given the number of datapoints, and how concordant many of the values are (like dAQ in A, for example), I wonder if these graphs actually include technical replicate values as well. Can the authors please clarify if this is the case.
- In Figure 2B, Drug_{in}/Drug_{out} referring to intracellular and extracellular accumulation should be specified in the figure legend.
- The range of values included in each column of the histograms in Figure 4B are unclear, especially for the isoelectric point. Perhaps the graph could be expanded to include more columns, or the ranges specified along the x-axis.

Responses to reviewers' comments on manuscript Spectrum01101-22

We thank the reviewers for their encouraging comments and helpful suggestions.

Reviewer comments:

Reviewer #1 (Comments for the Author):

Summary

The chloroquine resistance transporter of *Plasmodium falciparum* (PfCRT) has long been of significance to malaria parasite biologists for the role it plays in conferring resistance to the former frontline antimalarial chloroquine. Despite this, relatively little is known about the function of this transporter outside of its role in chloroquine transport. Moreover, most studies regarding CRT have used heterologous systems, making it difficult to extrapolate its *in situ* function(s) in malaria parasites. The authors utilise the GlmS ribozyme knockdown system to show that CRT is important for parasite growth and long-term fitness. Additionally, the authors show that CRT knockdown parasites become have increased chloroquine sensitivity and accumulation. Using both targeted and untargeted metabolomics, that authors show that the majority of the metabolites that are differentially detected when CRT is knocked down are small peptides, predominantly 2-4 amino acids long. Using microscopy, the authors then show that CRT is involved in the transport of small peptides out of the digestive vacuole, with CRT knockdown parasites exhibiting bloated digestive vacuoles and an accumulation of small peptides. Collectively, the authors robustly show that one of the functions of CRT in *P. falciparum* is the transport of small peptides. The conclusions drawn in this study are well supported by the data, and the authors should be commended for their attention to detail in-regards to statistical analysis and robustness. I have some comments, principally regarding the amount of detail provided in the methods sections about the experiments in this study, but I suspect these will be relatively easily addressed by the authors.

We thank the reviewer for the supportive comments.

Major comments:

- While the application of the GlmS knockdown system to Dd2 parental parasites has allowed the authors to study the function of CRT knockdown in a chloroquine-resistant parasite, the authors also note that the 8 mutations present in Dd2 CRT alter its activity. Considering this, the authors should mention in the discussion that it is currently unclear how similar the wildtype-functions of CRT are in chloroquine-sensitive and chloroquine-resistant parasites.

We agree with the reviewer and have included the following paragraph in the discussion:

“We acknowledge the limitation of our study in the sense that the findings are based on studies conducted with a *P. falciparum* strain that carries 8 mutations in its PfCRT protein. Efforts to generate a conditional PfCRT knock-down in a wild type strain have thus far failed in our laboratory. While it is undisputed that mutational changes in PfCRT can alter the substrate specificity of PfCRT, as has clearly been demonstrated for antimalarial drugs (12, 13), a variant-transcending role as an oligopeptide carrier is consistent with recent *in vitro* studies showing oligopeptide transport by both wild type and variant forms of PfCRT, including PfCRTDd2, albeit the transport capacity and the oligopeptide diversity were lower in drug resistance conferring PfCRT variants (29).” (page 12, second paragraph).

- Compared to the parental Dd2 parasites, E10 and G7 showed enlarged vacuoles and decreased fitness, even in the absence of glucosamine treatment. Because this is also the observed phenotype of CRT knockdown, this should be mentioned by the authors in the discussion, and if this is a phenomenon that has previously been observed using the GlmS system, the authors should discuss this too.

Several studies have used the glms system in *P. falciparum* and none have reported enlarged digestive vacuoles as a result of the genetic manipulation (Jankowska-Döllken *et al*, 2019; Prommana *et al*, 2013). We, therefore, have to assume an effect specific to PfCRT. Waller *et al*. have shown that manipulating the 3' UTR of the *pfcr*t gene can reduce *pfcr*t expression levels and drug responses (Waller *et al*, 2003). However, we do not see significant differences in PfCRT expression levels between the parental strain Dd2 and the two PfCRT knock-down mutants in the absence of glucosamine. Yet, the knock-down mutants have enlarged digestive vacuoles even in the absence of glucosamine. In the amended manuscript we speculate that this phenotype might result from the altered C-terminal end of PfCRT, which contains a hemagglutinin tag in the mutants, and a possible effect on the transport activity. We have added the following paragraph to the discussion to clarify this issue:

“It is further possible that the genetic alterations introduced in the *pfcr*t locus affected the transport efficiency of PfCRT. In this context, a previous study has shown that altering the 3' untranslated region of *pfcr*t decreased protein expression levels and tolerance to chloroquine (Waller *et al*, 2003). Although we observed comparable PfCRT protein levels in Dd2 and the PfCRT mutants in the absence of GlcN (Fig. 1C and D), we cannot exclude an effect of the C-terminally fused hemagglutinin tag on the transport activity of PfCRT in the inducible knock-down mutants. Such an effect might explain the already slightly enlarged digestive vacuoles in the mutants in the absence of GlcN (Fig. 5B). However, this putative limitation is not seen as an obstacle to the interpretation of our results and the conclusions drawn from them”. (page 12 third, third paragraph).

- In the methods, the authors should explain how the size of the digestive vacuole was measured. Additionally, the authors should provide details on how the fluorescence of the digestive vacuole was measured and how this was controlled.

We have added the following paragraph to explain the method in more detail:

“Images of infected erythrocytes stained with acridine orange (see above) were analyzed using Fiji and the area of the digestive vacuole was quantified. Assuming a spherical shape of the digestive vacuole (Saliba *et al*, 1998), its volume was calculated based on the following equation: $V = 4/3 \pi r^3$, where r is the radius.” (page 20, second paragraph).

- Further details should be provided on how the parasite growth assays were performed, including whether the parasites were synchronised and at what stage of the lifecycle glucosamine was added, currently it is unclear whether there are two rounds of invasion or 1 in Figure 1E.

We have amended the text as following:

“Highly synchronized parasite cultures at the ring stage (0.5% parasitemia) were split in two aliquots and one aliquot was incubated in the presence of 1mM GlcN for 72 h. The other aliquots served as the untreated controls. Parasite proliferation was assessed using Giemsa-stained thin

blood smears. Pyknic and crisis forms, as occurred in GlcN treated PfCRT knock-down parasite cultures, were not considered. The cumulative parasitemia was calculated as the sum of the parasitemia in the culture after considering splitting and dilution factors.” (page 15, third paragraph).

- Further details should be included about the parasite cultures used in Figure 2. Based on the details provided, it is currently unclear if, or how, the level of parasite growth inhibition caused by PfCRT knockdown could influence CQ and dAQ IC50 values.

The following paragraph was added to the Materials and Methods section:

“Cell proliferation assays to determine drug IC₅₀-values were performed as described (Sanchez *et al*, 2019), using the SYBR Green fluorescence-based assay (Smilkstein *et al*, 2004). Briefly, highly synchronized parasite cultures at the ring stage (0.5 % parasitemia, 3% hematocrit) were incubated in the presence and absence of glucosamine (1 mM) and increasing concentration of the drugs indicated. Cells were harvested 72 hours later and were lysed in lysis buffer (20 mM Tris-HCl, 5 mM EDTA, 0.008% saponin, 0.08% triton X-100, pH 7.4) containing SYBR green. Fluorescence was determined using a FLUOstar OPTIMA plate reader with the following settings: excitation wavelength: 485 nm; emission wavelength: 520 nm; gain: 1380; n° of flashes/well: 10; top optic.” (page 16, second paragraph).

- In Figure 6, the authors use a good control of VP to show that inhibition of CRT can lead to dipeptide accumulation, however, it seems there was no glucosamine control performed on Dd2 parasites. To rule out a possible influence of glucosamine on dipeptide accumulation, the authors should also include a Dd2 + GlcN control.

The requested Dd2 + GlcN control is now shown in the revised Figs. 6D (representative image) and 6E (quantification of the digestive vacuolar fluorescence signal). No specific accumulation of the fluorescently-labeled dipeptide VD was observed in the digestive vacuole of Dd2 in the presence of glucosamine. The new result is described on page 9, second paragraph.

Minor comments:

- Missing italics of species, gene names, or Latin in lines 58, 60, 100, 154.

We apologize for the oversights and have made the requested corrections.

- The authors often refer to the generated mutants as knockdown mutants, and the presence of glucosamine as the induced state. I found this a little confusing, particularly in the first two results paragraphs. It may be more clear to simply refer to them as in the presence or absence of glucosamine as this terminology can be applied to both Dd2 and the generated mutants.

We have changed the wording as suggested by the reviewer.

- Lines 118-120 refer to Dd2 in Figure 3, but it seems that only results from E10 and G7 are included in Figure 3.

We also conducted a metabolomics study on Dd2 in the presence and absence of GlcN. These data served as negative controls and were considered in the correlation analyses to determine the specific changes in the metabolome of the PfCRT knock-down mutants in the presence of GlcN (see Material and Methods section, page 19, end of metabolomics).

- It seems that the number of metabolites in Figure 4A and peptides in 4B are pooled from experiments of both E10 and G7. If this is the case, the authors should clarify this, or alternatively clarify if the results are only from a single clone

The reviewer is right. The data from the two independent PfCRT knock-down mutants were indeed pooled. We now state this in the result section (page 7, first paragraph). In this context, we would like to point out that the original data for each mutant and each condition are presented in the Tables S1 to S4.

- Line 180 says parasites were treated with 0.8 mm verapamil, but elsewhere in the text says 0.89 μm .

We apologize for the mistake and now correctly refer to 0.89 μM verapamil.

- Is there any particular reason the authors chose Val-Asp as the specific dipeptide? If so, this choice should be justified in the text.

The selection of the dipeptide Val-Asp was inspired by Shafik et al. (2020) who used VDPVNF for studies on PfCRT in the *X. laevis* oocyte system. We now state this in the Material and Methods section under Life cell imaging (page 20, first paragraph).

- The final paragraph of the results section does not include a reference to Supplementary Figure 1.

We apologize for the oversight and now refer to Figure S1 in the text.

- In the Methods, the subtitle that reads "Parasite culture and transfection" only mentions parasite culture, and transfection is covered in the next section.

The mistake has been corrected.

- The machine used to quantify SYBR green fluorescence should be listed in the methods section.

The section on "Drug response assays" has been re-written (see above) and the fluorimeter used in the study is now specified (page 16, second paragraph).

- The authors should provide further details on how the drug accumulation assay works, and list the machine used to measure the radiation from the label.

The section on the drug accumulation assay has been re-written and the requested detail is now provided (see Material and Methods section under "Drug response assays") (page 16, second paragraph).

- An Anti-POD antibody is listed in the Western blot section of the methods, but this antibody does not appear to have been used in this study.

The POD conjugated antibodies were used as secondary antibodies in the Western analysis shown in Figure 1C. This information is now provided in the figure legend and the Material and Methods section under "Western Analysis" (page 17, second paragraph).

- The legend of Figure 2 does not contain sufficient information to fully understand the figure. The authors should briefly explain how parasite growth and drug accumulation was measured for these assays.

The legend of Figure 2 has been improved to provide more experimental detail.

- From figure 6, it isn't clear what the function of verapamil is, this should be briefly listed in the legend.

We now state in the legend of Figure 6 that verapamil is a partial mixed type inhibitor of PfCRT^{Dd2} (Bellanca *et al*, 2014).

- In Figure 1A, there are a series of black lines along PfCRT that don't seem to be explained in the figure legend.

The vertical black lines represent introns, as now explained in the figure legend.

- In the fitness assay, none of the sequence proportions seem to drop significantly until 5 replicative cycles (~10 days). Yet in the growth assays, the glucosamine treated G7 and E10 parasites have only ~1/3 the parasitaemia of untreated after only 3 days. Can the authors provide any insight or justification on this discrepancy?

The mentioned discrepancy between the growth and the fitness assay regarding the diverging time lines is expected and results from the different experimental designs and read-out parameters of the two assays. The growth assay is based on microscopic inspection of Giemsa stained blood smears, with the read-out being health looking parasites. Pyknic and crisis forms and free parasites are usually not considered. In comparison, the fitness assay measures the relative presence of a certain DNA fragment, following amplification by PCR, in a co-culture set-up of two parasite lines. Thus, the fitness assay does not discriminate between different sources of the DNA and records DNA from healthy and dying parasites and, possibly, even free DNA. For this reason, the results from the fitness assay lag those of the growth assay, particularly during the first days following treatment with a drug or as in our case with GlcN when the culture still contains DNA from dying parasites. We clarified this issue in the Material and Methods section under "Asexual intraerythrocytic proliferation" and "Fitness assay" (page 15).

- The authors state in the figure legend that all datapoints in Figure 2 represent biological replicates. Given the number of datapoints, and how concordant many of the values are (like dAQ in A, for example), I wonder if these graphs actually include technical replicate values as well. Can the authors please clarify if this is the case?

All the data points are derived from biological replicates, as state in the figure legends and the Material and Methods section.

- In Figure 2B, Drugin/Drugout referring to intracellular and extracellular accumulation should be specified in the figure legend.

The legend of Figure 2B was amended to include the requested information.

- The range of values included in each column of the histograms in Figure 4B are unclear, especially for the isoelectric point. Perhaps the graph could be expanded to include more columns, or the ranges specified along the x-axis.

The ranges are now mentioned in the legend of Figure 4B.

Reviewer #2 (Comments for the Author):

This is an elegant, simple and well-presented work to assess the natural biological role of PfCRT. The data is clearly presented and supports very well the new proposed function of PfCRT in oligopeptide transport. The rigor of the results and the experimental design is high.

We thank the reviewer for the friendly and encouraging words.

Two minor suggestions:

1) Figure S1 does not really show a fitness cost. May be adding a growth curve or giemsa stained pictures will aid the readers in interpreting the results and conclusions from the authors.

As suggested by the reviewer we now show images of Giemsa-stained parasites to illustrate the observation that some of the fluorescently-labeled valine exerted some degree of toxicity to the parasite, as indicated by a general growth retardation and the presence of crisis forms (see Figure S1).

2) 5% albumax seems quite high as it is usually used at 0.5% wt/vol (or 5 g/L). please clarify the units.

We apologize for this mistake and have made the necessary correction in the text.

Reviewer #3 (Comments for the Author):

Title: The knock-down of the chloroquine resistance transporter PfCRT is linked to oligopeptide handling in Plasmodium falciparum

The study by Sanchez et al. shows that chloroquine resistance transporter PfCRT is involved in the transportation of the oligopeptides from the digestive vacuole of the parasite to the cytoplasm. These peptides are believed to be cleaved to single amino acids and are then recycled for biosynthesis purposes and/or expelled to maintain the colloid osmotic balance. This observation is in agreement with a recent study (Shafik et al., 2020, Nature Communications) as indicated by the authors. The quality of the data generated in this study is very good and it is well supported with statistics as and when required.

We thank the reviewer for the kind word.

However, I have a few major concerns in this study which are listed below:

1. The major premise of this work is that the non-drug related function of the PfCRT is less clear and the authors wanted to dissect the "Natural Function" of PfCRT. Also, as mentioned by the authors and known in the literature that several mutations acquired in the PfCRT affect its transporting property. This is also evident from the recent study by Shafik et al., 2020, Nature

Communications where the transporting properties of PfCRT in chloroquine-sensitive and resistant strains are distinct.

In this study, the authors have used the Dd2 strain, which is a chloroquine-resistant strain with eight polymorphisms that convert PfCRT to a transporter system that confers drug resistance. Thus, the effect observed in this study, are at best in comparison to the mutant PfCRT (mutations from the resistant strain). Therefore, to assign a physiological function to PfCRT, the authors should do these assays in a chloroquine-sensitive strain.

We fully agree with the reviewer. Ideally, one would repeat the work in a *P. falciparum* strain harboring a wild type *pfcr*t gene. We have tried this for several years, using the same approach as for the Dd2 strain, but failed. We have no rational explanation for this result and can only speculate about technical or biological reasons. To address the reviewers concern, we have included the following paragraph in the discussion:

“We acknowledge the limitation of our study in the sense that the findings are based on studies conducted with a *P. falciparum* strain that carries 8 mutations in its PfCRT protein. Efforts to generate a conditional PfCRT knock-down in a wild type strain have thus far failed in our laboratory. While it is undisputed that mutational changes in PfCRT can alter the substrate specificity of PfCRT, as has clearly been demonstrated for antimalarial drugs (12, 13), a variant-transcending role as an oligopeptide carrier is consistent with recent *in vitro* studies showing oligopeptide transport by both wild type and variant forms of PfCRT, including PfCRTDd2, albeit the transport capacity and the oligopeptide diversity were lower in drug resistance conferring PfCRT variants (29).” (page 12, second paragraph).

2. Figure 1 C and D: Even though a decrease is observed upon induction with glucosamine, there seems to be a significant increase in PfCRT expression in the uninduced knockdown parasite lines (E10 and G7) as compared to the Dd2 parent line. It is not clear if the tagging of the endogenous locus leads to this effect and in turn may have an effect on the transporting property of PfCRT. The authors need to clarify this effect.

This is a misperception. There are no significant differences in the steady-state amount of PfCRT between Dd2 and the PfCRT knock-down mutants in the absence of GlcN, as demonstrated by at least six biologically-independent semi-quantitative Western blots and normalization of PfCRT-specific signals to that of α -tubulin ($p > 0.8$; Kruskal-Wallis one way ANOVA on ranks).

3. Authors can also include a brief explanation on the accumulation and localization of specifically dipeptides in the digestive vacuole in Dd2 and conditional knockdown lines with respect to chloroquine resistance. Do authors expect this to be the same in a chloroquine-sensitive line?

We now discuss our results in the context of the chloroquine resistant background of the *P. falciparum* lines used in this study (see comment above).

Additional references

Bellanca *et al.* (2014) Multiple drugs compete for transport via the Plasmodium falciparum chloroquine resistance transporter at distinct but interdependent sites. *The Journal of biological chemistry* 289: 36336-36351

Jankowska-Döllken *et al.* (2019) Overexpression of the HECT ubiquitin ligase PfUT prolongs the intraerythrocytic cycle and reduces invasion efficiency of Plasmodium falciparum. *Scientific reports* 9: 18333

Prommana *et al.* (2013) Inducible knockdown of Plasmodium gene expression using the glmS ribozyme. *PLoS One* 8: e73783

Saliba *et al.* (1998) Role for the Plasmodium falciparum digestive vacuole in chloroquine resistance. *Biochemical pharmacology* 56: 313-320

Sanchez *et al.* (2019) Phosphomimetic substitution at Ser-33 of the chloroquine resistance transporter PfCRT reconstitutes drug responses in Plasmodium falciparum. *The Journal of biological chemistry* 294: 12766-12778

Smilkstein *et al.* (2004) Simple and inexpensive fluorescence-based technique for high-throughput antimalarial drug screening. *Antimicrobial agents and chemotherapy* 48: 1803-1806

Waller *et al.* (2003) Chloroquine resistance modulated in vitro by expression levels of the Plasmodium falciparum chloroquine resistance transporter. *The Journal of biological chemistry* 278: 33593-33601

June 27, 2022

Prof. Michael Lanzer
Heidelberg University
Center of Infectious Diseases, Parasitology
Heidelberg D-69120
Germany

Re: Spectrum01101-22R1 (The knock-down of the chloroquine resistance transporter PfCRT is linked to oligopeptide handling in *Plasmodium falciparum*)

Dear Prof. Michael Lanzer:

Your manuscript has been accepted, and I am forwarding it to the ASM Journals Department for publication. You will be notified when your proofs are ready to be viewed.

Sincerely,

Anat Florentin
Editor, Microbiology Spectrum
